# [Re] Key Point Analysis via Contrastive Learning and Extractive Argument Summarization

## Reproducibility Summary

**Scope of Reproducibility**

The goal of this work is to validate the reproducibility of key point analysis of arguments framework proposed by (1). The authors claimed that they achieved the best performance in the KPA shared task via contrastive learning. For key point generation, they developed a graph-based extractive summarization model that output informative key points of high quality for a collection of arguments.

**Methodology**

We used open source code of the authors with slight changes. Simple parts of code were run on CPU, while the parts that require training and working with deep models were run on a NVIDIA Tesla K80 GPU (with 12GB memory, which is the Google Colab's [1] default GPU) for 2 hours approximately.

**Results**

1. We reproduced the results of paper on the provided test set with the following details:

   - Our ROUGE-1 metric was within 0.99% of the reported value which is acceptable.
   - Our ROUGE-2 metric was within 7.14% of the reported value which is a little high.
   - Our ROUGE-L metric was within 1.07% of the reported value which is acceptable.

   Check notebook number 6 [2] for this part of results.

2. There are also some metrics for evaluation of key point matching on validation set with the following details:

   - Our strict mAP on the validation set was the same as the reported value (with accuracy of one hundredth of a decimal) and relaxed mAP metric was within 1.04% of the reported value which is acceptable.

   Check notebook number 3 [3] for this part of results.

It can be said that the results of reproduction were generally acceptable.

**What was easy**

It was easy to run and config most parts of the provided code in the repository of the paper, except some parts that we will cover in the next session.

---

[1] www.colab.research.google.com

[2] www.anonymous.4open.science/r/argmining-21-keypoint-analysis-sharedtask-code-554D/code/src-ipynb/6.experiment$_e valuation.ipynb$

[3] www.anonymous.4open.science/r/argmining-21-keypoint-analysis-sharedtask-code-554D/code/src-ipynb/3.experiment-evaluation.ipynb

## What was difficult

Some parts of code like the notebook number 4 [4] in our repository was unable to run because of timeout errors, which was easy to solve by some exception handling. Furthermore, matching the datasets, because of having two groups of data and having some extra data which was not used in the code, was a little hard.

## Communication with original authors

The official implementation is complicated thus not easy to follow. We contacted the first author about the order of running files so the author cleaned the git repository of code but some of files were missing that were available from previous commit.

---

[4]www.anonymous.4open.science/r/argmining-21-keypoint-analysis-sharedtask-code-554D/code/src-ipynb/4.experiment-data-prep-for-track-2.ipynb

# 1 Introduction

Search engines benefit from employing argument summarization, that is, the generated summaries may aid the decisionmaking by helping users quickly choose relevant arguments with a specific stance towards the topic. Argument summarization has been investigated in single documents (2) and multiple documents (3).

(4) introduced key point analysis that is the task of extracting a set of concise and high-level statements from a given collection of arguments, representing the gist of these arguments. The original paper presented an approach with two complementary subtasks: matching arguments to key points and generating key points from a given set of arguments. We expained each subtask in Section 3.

# 2 Scope of reproducibility

Beyond the scope of the original paper. The main claim of the original paper is:

- The graph-based summary provides a more comprehensive overview than aspect clustering.

# 3 Methodology

## 3.1 Model descriptions

the KPA shared task consists of two subtasks as described below:

- Key point matching. Given a set of arguments on a certain topic that are grouped by their stance and a set of key points, assign each argument to a key point.
- Key point generation and matching. Given a set of arguments on a certain topic that are grouped by their stance, first generate five to ten key points summarizing the arguments. Then, match each argument in the set to the generated key points (as in the previous track).

For Key point matching the original proposed a model that learns a semantic embedding space where pairs of key point and argument that match are closer to each other while non-matching pairs are further away from each other. They embed pairs by utilizing a contrastive loss function in a siamese neural network (5). They computed the contrastive loss using output embeddings siamese neural network of as follows:

$$\mathcal{L} = -y \log \hat{y} + (1 - y) \log (1 - \hat{y}) \tag{1}$$

where $\hat{y}$ is the cosine similarity of the embeddings, and y reflects whether a pair matches (1) or not (0).

For Key point generation the paper proposed a primary model that is a graph-based extractive summarization model. Additionally, they also investigate clustering the aspects of the given collection of arguments.

**Graph-based Summarization:**
In this method they first constructed an undirected graph with the arguments' sentences as nodes and exclude low-quality arguments from the graph with argument quality scores introduced by (6). Next the key point matching model was employed to compute edge weights between two nodes. Only nodes with a score above a defined threshold are connected. Finally a variant of PageRank (7) was used to compute importance score $P(s_i)$ for each sentence $s_i$ as follows:

$$P(s_i) = (1 - d) \sum_{s_j \neq s_i} \frac{match(s_i, s_j)}{\sum_{s_k \neq s_j} match(s_j, s_k)} P(s_j) + d \frac{qual(s_i)}{\sum_{s_k} qual(s_k)} \tag{2}$$

where d is a damping factor.To ensure diversity, the method iterates through the ranked list of sentences (in descending order), adding a sentence to the final set of key points if its maximum matching score with the already selected candidates is below a certain threshold.

**Aspect Clustering:**
Extracting key points is similar to identifying aspects (4) and selects representative sentences from multiple aspect

clusters as the final key points. The tagger of (8) was employed to to extract the arguments' aspects (on average, 2.1 aspects per argument). At the end they tackled the lack of diversity and avoided redundant key points concurrently.

## 3.2 Datasets

All datasets are available in the author's repository [5]. The information about datasets is described in the following:

Train dataset:

- Number of samples: 20635
- Features name: arg_id, key_point_id, argument, topic, stance, key_point

We can see two histograms about key points and arguments as the following:

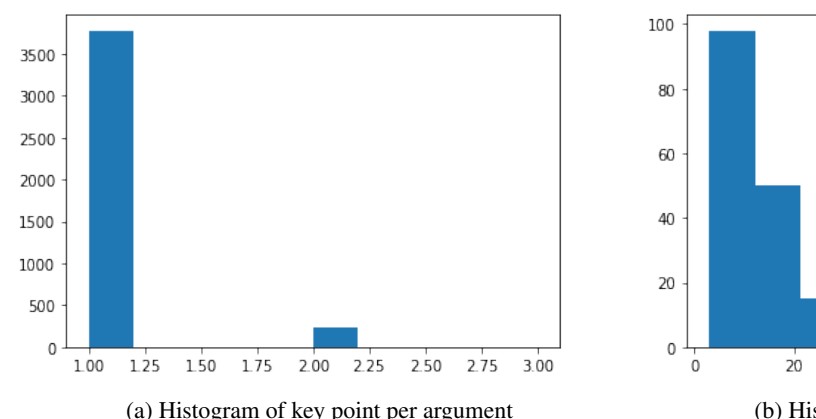

(a) Histogram of key point per argument      (b) Histogram of argument per key point

Figure 1: Useful statistics about train set

Validation dataset:

- Number of samples: 2400
- Features name: arg_id, key_point_id, argument, topic, stance, key_point
- Number of unique argument: 653
- Number of unique key point: 36

Test dataset:

- Number of samples: 1058
- Features name: arg_id, key_point_id, argument, topic, stance, key_point
- Number of unique argument: 279
- Number of unique key point: 36

## 3.3 Hyperparameters

For key point matching these hyperparameters were used:

- Number of epochs: 10
- Batch size: 32
- Maximum input length: 70
- All other parameters are left to their defaults.

---

[5]www.anonymous.4open.science/r/argmining-21-keypoint-analysis-sharedtask-code-554D/data

For key point generation these hyperparameters were used:

- Sentences length are between 5 and 20 tokens.

- d, qual and match in Equation 2 are selected as 0.2, 0.8 and 0.4 respectively.

- ROUGE-L between the ground-truth key points and the top 10 ranked sentences are computed as predictions.

- Sentences with a matching score higher than 0.8 with the selected candidates are excluded to minimize redundancy.

There is no info about searching for hyperparameters in the paper.

## 3.4 Experimental setup and code

Codes from the author's repository [6] were forked and with little changes and some comments are available at our repository repository [7].

The note books are easy to run by considering the comments and order of execution which is the same as the prefix number in notebook's name. It is better to run notebooks on Google Colab except the notebook number 4 (which was mentioned in What was difficult section of report) because it needs approximately long time for execution and disconnects from Google Colab's runtime and also raises network timeout error.

Strict and relaxed mAP (mean Average Precision) (9) are used for automatic evaluation. In cases where there is no majority label for matching, the relaxed mAP considers them to be a match while the strict mAP considers them as not matching (1).

ROUGE-1, ROUGE-2, ROUGE-L metrics (10) are used for key point generation evaluation. The formula of the metrics are available at the cited paper. Here we only define metrics briefly (11):

- ROUGE-1 refers to the overlap of unigram (each word) between the system and reference summaries.

- ROUGE-2 refers to the overlap of bigrams between the system and reference summaries.

- ROUGE-L: Longest Common Subsequence (LCS)[3] based statistics. Longest common subsequence problem takes into account sentence level structure similarity naturally and identifies longest co-occurring in sequence n-grams automatically.

## 3.5 Computational requirements

At the top of each notebook it has been noted to use GPU or not. Notebook number 4 can be run on any simple system, because it needs to request to API. To receive API keys you should read this [8] link and follow the instructions.

All the notebooks use NVIDIA Tesla K80 GPU (with 12GB memory, which is the Google Colab's default GPU).

- Notebook 2 needs at about 1 hour and 20 minutes for training.

- Notebook 3 takes a few minutes to run (too short to consider).

- Notebook 5 takes at about 30 minutes to run.

- Notebook 6 takes a few minutes to run (too short to consider).

- Other notebooks (1, 4) do not use GPU.

## 4 Results

As we said before results were acceptably reproduced the paper's main results. We will see the results with more details in the following section.

---

[6] www.github.com/webis-de/argmining-21-keypoint-analysis-sharedtask-code
[7] www.anonymous.4open.science/r/argmining-21-keypoint-analysis-sharedtask-code-554D/README.md
[8] www.early-access-program.debater.res.ibm.com

### 4.1 Results reproducing original paper

Results for two main parts (key point matching and key point generations) are provided in this section.

#### 4.1.1 Result for key point matching

There are also some metrics for evaluation of key point matching on validation set with the following details:

- Our strict and relaxed mAP on the validation set were 0.84 and 0.97 respectively while the reported strict and relaxed mAP on the validation set were 0.84 and 0.96 respectively.
  We see that strict mAP is approximately same for both experiments and relaxed mAP has a difference of 1.0.4%.

Check notebook number 3 [9] for this part of results.

Results of this part are available in Table 1 1.

| metrics | reproduced | reported |
|---|---|---|
| strict mAP | 0.84 | 0.84 |
| relaxed mAP | 0.97 | 0.96 |

Table 1: key point matching results.

#### 4.1.2 Result for key point generation

For key point generation we reproduced the results of paper on the provided test set with the following details:

- Our ROUGE-1 metric was 0.204 while the reported ROUGE-1 metric was 0.202 which the difference was about 0.99% which is acceptable.
- Our ROUGE-2 metric was 0.039 while the reported ROUGE-2 metric was 0.042 which the difference was about 7.14% which is a little high.
- Our ROUGE-L metric was 0.188 while the reported ROUGE-L metric was 0.186 which the difference was about 1.07% which is acceptable.

Check notebook number 6 [10] for this part of results.

Results of this part are available in Table 2 2.

| metrics | reproduced | reported |
|---|---|---|
| ROUGE-1 | 0.204 | 0.202 |
| ROUGE-2 | 0.039 | 0.042 |
| ROUGE-L | 0.188 | 0.186 |

Table 2: key point generation results.

## 5 Discussion

After evaluation of the framework on the provided datasets in the repository, almost all part of the results were reproduced acceptably, except the claim that says the graph-based summary provides a more comprehensive overview than aspect clustering. The piece of code for reproducing this claim was not found at the last commit of the provided code in the repository. But we know that last commit was a fast refactoring of the code, so some notebooks might be missing and might be found in the previous commits. We had not enough time to look for it and it was a little confusing.

---

[9] www.anonymous.4open.science/r/argmining-21-keypoint-analysis-sharedtask-code-554D/code/src-ipynb/3.experiment-evaluation.ipynb

[10] www.anonymous.4open.science/r/argmining-21-keypoint-analysis-sharedtask-code-554D/code/src-ipynb/6.experiment$_{evaluation.ipynb}$

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
