# OpenReview forum: "[Re] Key Point Analysis via Contrastive Learning and Extractive Argument Summarization"
_ML_Reproducibility_Challenge/2021/Fall — Reject_

### Official Review · Reviewer_8Hus · 2022-03-17
**Keypoint Generation Seems Not Reproducible**

**Rating:** 5
**Confidence:** 3

**Review:**

This work aims to reproduce the work "Key Point Analysis via Contrastive Learning and Extractive Argument Summarization". Based on the provided notebooks, this work was able to reproduce most of the results claimed in the original paper, except R-2 in keypoint generation.

Pros:
1. This work is able to run most of the experiments in the original paper with few exceptions.
2. This work found that keypoint generation is not reproducible and there is a big difference in R-2, which is might be a useful information for followup works of the original paper.

Cons:
1. The most outstanding issue is that this work was unable to verify "the claim that says the graph-based summary provides a more comprehensive overview than aspect clustering" (L155), which seems to be a major claim of the original paper. That being said, it is useful to know that the code for verifying this claim is not in the current commit of the released repo.
2. This work mentioned slight changes were made to the original code. I think it would be nice to elaborate why these changes are necessary (e.g., do you encounter errors running the original code?)
3. I think the difference in R-2 should be highlighted instead of concluding that "It can be said that the results of reproduction were generally acceptable."
4. There are some typos and presentation issues.

Presentation Issues:
1. For tables, I recommend not using vertical lines, and use \toprule and \bottomrule from the latex library booktabs.
2. L52 For Key point matching the original proposed a model -> original paper
3. L68 actor.To  -> space missing
4. L107 note books -> notebooks

Overall, this work is able to prove/disprove many results of the original paper. However, a major claim of the original paper (see con #1) is not checked. Therefore, I think this work is slightly below the acceptance threshold.

---

### Official Review · Reviewer_6M5c · 2022-03-30
**The report was mere re-running of the original code without any hyper-parameter search or any other ablation studies. The paper missed the very motivation of reproducbility.**

**Rating:** 2
**Confidence:** 2

**Review:**

Major Comment: The original paper venue does not lie within the scope of accepted venues at MLRC2021.

Other Comments
- Reproducibility Summary: Provided
- Scope of reproducibility: Provided
- Code: Re-used author's repository. A readme is present which points to various code notebooks but other than this no other doc is provided. Code is also not commented on properly.
- Communication with original authors: Provided but unsatisfactory.

Feedback from the reviewer:
The focus of the report was on rerunning the original author's code, and lacks basic motive of reproducibility.

- Hyperparameter Search: *NO* hyperparameter search performed.
- Ablation Study: *NO* ablation study done
- Presentation: A large scope of improvement.
- Results beyond the paper: *NOT* Provided

---

### Meta-Review · Program_Chairs · 2022-04-07

**Recommendation:** Reject
**Confidence:** 5

**Metareview:**

While the paper is well-structured and provides some relevant contributions, there are numerous missing checks in this paper (notably, a verification of the main claim from the paper that is being reproduced), which could be improved with further experimentation.

---

### Decision · Program_Chairs · 2022-04-09

Reject